# Understanding the Role and Clinical Applications of Exosomes in Gynecologic Malignancies: A Review of the Current Literature

**DOI:** 10.3390/cancers14010158

**Published:** 2021-12-29

**Authors:** Molly Roy, Yu-Ping Yang, Olivia Bosquet, Sapna K. Deo, Sylvia Daunert

**Affiliations:** 1Sylvester Comprehensive Cancer Center, Division of Gynecologic Oncology, University of Miami, Miami, FL 33136, USA; yyang22@med.miami.edu; 2Department of Biochemistry and Molecular Biology, University of Miami, Miami, FL 33136, USA; oxb179@miami.edu (O.B.); sdeo@med.miami.edu (S.K.D.); 3Dr. John T. Macdonald Foundation Biomedical Nanotechnology Institute, University of Miami, Miami, FL 33136, USA; 4Miami Clinical and Translational Science Institute, University of Miami, Miami, FL 33136, USA

**Keywords:** tumor-derived exosomes, diagnostics, therapeutics, gynecologic malignancies

## Abstract

**Simple Summary:**

Gynecologic malignancies are those that affect the female reproductive organs, including the ovary, uterus, and cervix. Gynecologic malignancies as a group vary significantly in initial clinical presentation, degree and pattern of spread, and treatment modalities. Our knowledge of the role exosomes play in cancer growth and spread is expanding rapidly. The promise these nanovesicles hold in cancer diagnostics and therapeutics is undeniable, and may be the key to improving the diagnosis and treatment of gynecologic cancers. This paper serves to review laboratory techniques utilized in isolating and studying exosomes, the current state of understanding of exosomes in gynecologic malignancies, and the potential for clinical applications that exosomes may hold.

**Abstract:**

Background: Gynecologic malignancies are those which arise in the female reproductive organs of the ovaries, cervix, and uterus. They carry a great deal of morbidity and mortality for patients, largely due to challenges in diagnosis and treatment of these cancers. Although advances in technology and understanding of these diseases have greatly improved diagnosis, treatment, and ultimately survival for patients with gynecologic malignancies over the last few decades, there is still room for improvements in diagnosis and treatment, for which exosomes may be the key. This paper reviews the current knowledge regarding gynecologic tumor derived-exosomal genetic material and proteins, their role in cancer progression, and their potential for advancing the clinical care of patients with gynecologic cancers through novel diagnostics and therapeutics. Literature Review: Ovarian tumor derived exosome specific proteins are reviewed in detail, discussing their role in ovarian cancer metastasis. The key microRNAs in cervical cancer and their implications in future clinical use are discussed. Additionally, uterine cancer-associated fibroblast (CAF)-derived exosomes which may promote endometrial cancer cell migration and invasion through a specific miR-148b are reviewed. The various laboratory techniques and commercial kits for the isolation of exosomes to allow for their clinical utilization are described as well. Conclusion: Exosomes may be the key to solving many unanswered questions, and closing the gaps so as to improve the outcomes of patients with gynecologic cancers around the world. The potential utilization of the current knowledge of exosomes, as they relate to gynecologic cancers, to advance the field and bridge the gaps in diagnostics and therapeutics highlight the promising future of exosomes in gynecologic malignancies.

## 1. Introduction

The presence of extracellular vesicles (EVs) was first described in 1946, and their biological contribution in cellular communication in the 1980s [1]. Since then, extracellular vesicles have been better understood further classified into multiple groups, one of which is exosomes. Other types of extracellular vesicles similar in size and physical properties to exosomes include apoptotic bodies, and ectosomes [2]. The main difference between these EVs is their mode of biogenesis, which determines the contents and function of each of the vesicles [2]. Exosomes are the smallest of the nanosized extracellular vesicles, typically ranging from 30–100 nm [2]. Exosomal biogenesis and the various exosomal cargos are illustrated in Figure 1.

While non-exosomal EVs originate from the direct budding of plasma membranes, exosomes are the result of the inward budding of endosomes into multivesicular bodies (MVBs). Some MVBs are sorted to form intraluminal vesicles, which contain information from the parent cell in the form lipids, proteins, and nucleic acids [3]. For this reason, exosomes excreted into the extracellular fluid carry functional proteins that vary depending on their origin, and their contents directly reflect the metabolic state of their host cells [4]. There are, however, some similarities among all exosomes, regardless of their host cells. All exosomes share features of certain tetraspanin (CD9, CD63, and CD81), heat shock proteins (Hsp 60, Hsp 70, and Hsp 90), and biogenesis-related proteins (Alix and TSG 101) [3]. It is now understood that the content of the exosomes from the parent cells allows signals to be transmitted to neighboring and distant cells without direct contact [5]. While the role of exosomes in cell-to-cell communication has been of interest in many aspects of cellular biology, it is this relationship to the parent cell that also makes exosomes of particular importance in cancer biology.

Clinically, most cancer cells are understood to be spread largely lymphatically or hematologically. Further, it has been observed that certain tumor types metastasize preferentially to certain organs. Recent studies have shown that exosomes are an integral part of tumor metastasis. For example, it has been demonstrated that exosomes from breast cancer cells could transfer invasion-promoting properties to tumorigenic, but not metastatic breast cancer cells. Further, it has been shown that exosomes from the primary tumor cells in pancreatic cancer and melanoma create a metastatic niche in distal tissue. That is, exosomes modify the extracellular matrix of lymph node or endothelial stromal cells to promote migration and proliferation, and to resist apoptosis [5,6].

Understanding these properties of cancer growth and spread has opened the door to many diagnostic and therapeutic applications of exosomes. The relationship exosomes have to their parent cells allow them to serve as biomarkers in some cancers. Notably, cancer cells have been shown to release exosomes at a higher level than other cells, and their contents remain well protected within their lipid bilayer membranes. In both small cell lung cancer and breast cancer, proteins identified in circulating exosomes have been identified as biomarkers [7]. The contents of exosomes are becoming increasingly better understood, as is their role in promoting tumor proliferation and metastasis. Therefore, targeting exosomes has become a focus of cancer therapeutics. In melanoma, blocking the internalization of tumor derived exosomes has been shown to prevent changes in normal cells to favor the tumor microenvironment [8]. Further, the use of exosomes as gene delivery or drug delivery vehicles is also being explored.

Our knowledge of the role exosomes play in cancer growth and spread is expanding rapidly. The promise these nanovesicles hold in cancer diagnostics and therapeutics is undeniable. Gynecologic malignancies are those that affect the female reproductive organs. Most commonly, these include the ovary, uterus, and cervix. This paper serves to review laboratory techniques utilized in isolating and studying exosomes, the current state of understanding of exosomes in gynecologic malignancies, and their potential diagnostic and therapeutic clinical applications.

## 2. Identified Roles of Exosomes in Gynecologic Malignancies

### 2.1. Ovarian Cancer

Ovarian cancer is the most common cause of death from gynecological malignancies in the world. The clinical challenges in treating ovarian cancer are that it is often diagnosed in a late stage and that it recurs rapidly after initial treatment. For this reason, there is much interest in developing methods of early diagnosis, detecting treatment response early, and targeting therapies in ovarian cancer. There have been recent strides in utilizing exosomes to do so. Approximately 2000 species of protein have been identified in exosomes derived from ovarian cancer cells. Many of these proteins have been shown to be involved in tumor metastasis and progression, and were associated with carcinogenesis through highly enriched signal pathways [4]. Tumor-derived exosomes have been identified in peripheral blood as well as ascitic fluid [4]. As identified in other types of malignancy, tumor metastasis is supported by the exosomal mediation of epithelial-to-mesenchymal transition, migration, invasion, angiogenesis, immune modulation and metabolic, epigenetic, and stromal reprogramming to create a pre-metastatic niche.

A characteristic of ovarian cancer metastasis is peritoneal dissemination, in which cancer cells detach from the primary bulk tumor and spread through the peritoneal cavity, attaching to other peritoneal organs. Exosome-mediated crosstalk between cancer cells and the tumor microenvironment is deeply involved in each step of peritoneal dissemination. Exosomal-mediated communication between the cancer cells and tumor microenvironment is thought to be integral to each step of dissemination. Exosomes containing L1 adhesion molecules (CD171) have been identified in the ascitic fluid in ovarian cancer, and have been shown to trigger cell migration. Hypoxic stress to the floating cancer cells induce increased exosome production. These exosomes are enriched in phosphorylated STAT3, which promotes invasion into the peritoneal surface. Additionally, the biologic content of the exosomes found in ascitic fluid contribute to creating a pre-metastatic niche [9], which is an essential step in cell invasion and proliferation, allowing for ovarian cancer dissemination.

Exosome protein signatures specific to those derived from ovarian cancer have been identified, which function to create a metastatic niche. One such protein, Nanog, a transcription regulator important in tumor cell proliferation and cancer stem cell self-renewal, has been detected in significantly greater numbers in exosomes found in ascites from ovarian cancer [10]. A number of other proteins, such as membrane proteins, Rab proteins, annexin proteins, heat shock proteins, and tetraspanins have also been discovered to be integral in creating the pre-metastatic niche. In addition to creating a supportive microenvironment for metastasis, ovarian cancer-derived exosomes have also been shown to cause immunosuppression within the niche. Exosomes isolated from ascites can induce T cell arrest [10]. Furthermore, ascitic exosomes carry activating transcription factor 2 (ATF2) and metastasis-associated protein 1 (MTA1), both of which promote angiogenesis in the metastatic environment [11]. Stromal cell remodeling to allow for tumor cell survival also occurs as a result of ovarian cancer-derived exosomes [10]. The vital role exosomes play in cancer cell survival and growth is becoming abundantly clear, opening the door for future interventions utilizing exosomes to regulate tumor growth and metastasis.

### 2.2. Cervical Cancer

Cervical cancer is the second most common cancer in the world for women. Human Papillomavirus (HPV) has been shown to be the main cause of cervical squamous cell carcinoma, and has been a focus of cervical cancer diagnostic and therapeutic research over the past few decades [3]. More recent developments in cancer biology have revealed that exosomes play an important role in cervical cancer growth. In contrast to the metastatic pattern of distant peritoneal dissemination common to ovarian cancer, cervical cancer progression occurs initially in the form of local expansion, although lymph node and distant tissue metastasis also occurs. Despite this difference, the principles of creating and maintaining a tumor microenvironment supportive of tumor growth are universal. Therefore, similar to proteins identified in the exosomes from ovarian cancer, the cargo of exosomes derived from cervical cancer are key for intra-cellular communication promoting tumor growth.

Local angiogenesis has been shown to be essential for cervical tumor growth [12]. Communication via cervical cancer-derived exosomes is now recognized as important in promoting angiogenesis. Exosomes are known to transport functional RNAs from cancer cells to stromal cells [13]. One such RNA is the microRNA (miRNA) miR-221-3p, which has been identified to play a key role in both angiogenesis and lymphangiogenesis [12,14]. MiR-221-3p is found in higher concentrations in cervical cancer cells, and is enriched in cervical cancer-derived exosomes [12]. miR-221-3p is transported in exosomes to epithelial cells, and it promotes angiogenesis, thereby promoting tumor growth [12]. Likewise, exosomes transport miR-221-3p to lymphatic endothelial cells, which facilitates lymph node metastasis [14]. Additional miRNAs and proteins found in exosomes continue to be explored, but from this one example, the large contribution of exosomes in supporting tumor expansion can be appreciated.

Solid tumors, like cervical cancer, are understood to have an organ-like nature. This is supported by tumor innervation as a component of tumor progression, as was extrapolated from head and neck squamous cell carcinomas [15]. Interestingly, exosomes derived from HPV-negative tumors mediated cellular changes that allowed neurite outgrowth. The loco-regional spread of these exosomes interacts with the nearby nerves to stimulate their extension into the tumor parenchyma [15]. While at this time, the prognostic significance of tumor innervation is not clearly understood in cervical cancer, is has been correlated with worse prognosis in other solid tumors [15]. Additional studies to understand the role of exosomes in innervation, especially in HPV-negative cervical cancers, hold much clinical promise.

### 2.3. Uterine Cancer

Uterine cancer is the most common gynecologic malignancy in the United States [16]. As in other malignancies, the tumor microenvironment is vital in determining how tumor cells grow and spread. In uterine cancer, cancer-associated fibroblasts (CAF) have been identified as a key component in the tumor microenvironment that promote cancer cell development, migration, and invasion [17]. Exosomes play an integral role in cell-to-cell communication between CAFs and the tumor cells, such that CAF-derived exosomes may promote endometrial cancer cell migration and invasion [18]. This, as demonstrated in cervical cancer, appears to be through miRNA. Specifically, miR-148b has been identified as an inhibitor of endometrial cancer growth, and is noted to be in lower concentrations in CAFs, and therefore it is transferred less in exosomes to tumor cells. These decreased levels contribute to increased tumor growth in endometrial cancer cells [18]. Knowing that CAFs are an integral part of the uterine tumor environment and that exosomes are key in cellular communications, there are likely to be a number of additional bioactive molecules identified in uterine cancer exosomes contributing to tumor spread.

Figure 2 illustrates the spread of gynecologic tumor derived-exosomes contributing to tumor metastasis in the 3 most common gynecologic malignancies.

## 3. Lab Techniques to Isolate Exosomes

The clinical promise of exosomes is becoming clearer with increasing understanding. Tumor-derived exosomes are present in almost all biological fluids such as blood, plasma, saliva, ascites, vaginal secretions, and urine [4]. With the perfection of isolation techniques, the utilization of exosomes in diagnostic or therapeutic clinical settings is closer to becoming a reality. There are various techniques for the isolation of exosomes, each of which are based on a specific principle and carry their own unique benefits and challenges. These techniques include those based on size, density, precipitation, immunoaffinity, and microfluidics. There are also certain commercially available kits for exosome isolation that can be purchased and used in cancer research. The common isolation techniques are illustrated in Figure 3 and the benefits and challenges for each are summarized in Table 1.

### 3.1. Density-Based Ultracentrifugation Techniques

Differential Centrifugation. Differential centrifugation was one of the first methods used for exosome isolation, and continues to be one of the most widely used [19]. In this method, exosomes are isolated from other particles in the sample based on their density and size via a series of centrifugation cycles of differing centrifugal force and duration [20]. Typically, the first rounds of centrifugation, which occur between 300× *g* and 1000× *g*, remove dead cells and cell debris. A round of centrifugation at around 10,000× *g* is performed to remove apoptotic bodies and larger extracellular vesicles (EV) [20]. Ultracentrifugation, which occurs at around 100,000× *g*, then isolates the pellet containing the isolated exosomes, which are then resuspended and stored at −80 °C for further analysis [7]. It has been shown that a longer spin duration at 100,000× *g* can increase the exosomal yield, however a spin time of greater than 4 h can result in damage to the exosome and contamination by soluble proteins [21]. Some benefits of this method include that it is fairly easy to use, requiring little technical expertise, and that it can isolate exosomes from a fairly large sample [22,23]. However, this technique results in a fairly low exosome yield and recovery rate, and it is incredibly time consuming [7]. Additionally, centrifugation time, force, and temperature can alter the structure of the recovered exosome and potentially result in damage to the exosome [23].

Density Gradient Centrifugation. Density gradient centrifugation is based on the same separation principles as differential centrifugation, but this separation occurs with the use of a preconstructed density gradient medium in the centrifuge tube [19,20]. The sample, containing exosomes and other particles, is layered at the top of the density gradient and, upon application of centrifugal force, moves through the density gradient at distinct rates that allow for separation of the particles. The exosomes can then be collected by fractionation collection [19,22]. Density gradient centrifugation allows for higher purity exosomes than the differential centrifugation technique. However, capacity is limited to a narrow loading area [7,19]. Additionally, this technique shares some of the same limitations and differential centrifugation, especially in that it is time-consuming and may alter the exosome structure [23].

### 3.2. Size-Based Techniques

Size-Exclusion Chromatography (SEC). Size-exclusion chromatography (SEC) separates exosomes from other extracellular vesicles on the basis of their size. The sample containing the exosomes is passed through a column containing a porous stationary phase of a particular size that is penetrable by smaller particles [7,19]. As the sample passes through the column the larger particles, which cannot penetrate the pores, flow quickly through the column and elute first while the smaller particles, which remain in the pores, are eluted later with the use of a mobile phase [19,24]. This method may be used in conjunction with ultracentrifugation methods, in which the exosomes are enriched by centrifugation, then resuspended and purified using SEC [19]. The benefits of SEC include its ability to maintain the structural integrity and biological activity of the exosomes, its relative low cost, the high purity of the isolated exosomes, and the lack of albumin contamination [23,24]. However, this method can only accommodate a small sample volume and results in a low exosome yield [23].

Ultrafiltration. The method of ultrafiltration involves the use of a nanoporous filtration membrane of a particular size and molecular weight cutoff (MWCO), generally between 0.1 and 0.001 µm [7,19]. These filters can separate the bioparticles based on their size so that particles larger than the cutoffs are retained by the filter, while the remaining particles flow through it [19]. This technique may also be combined with ultracentrifugation techniques, in that some of the lower speed centrifugation rounds are replaced with ultrafiltration [7]. Compared with ultracentrifugation methods, ultrafiltration has shown higher exosome purity and is somewhat less time consuming [25]. Some limitations of this technique include that exosome structure deformation may occur due to the use of force, and that non-specific binding and trapping of the exosomes to the membrane may occur, resulting in a reduced recovery rate [7,19,24].

Sequential Filtration. Sequential filtration relies on a series of filtration steps for exosome enrichment [19]. In the initial filtration, a 100 nm filter is used to eliminate large rigid particles, including cells and cell debris [19,22]. Particles that are large but flexible, including exosomes, are able to pass through the filter [19,22]. The filtrate then undergoes tangential flow filtration, using a 500 kDa MWCO membrane, which removes soluble proteins and other contaminants [19]. Lastly, the sample is filtered through a 100 nm track-etch filter for exosome enrichment [19]. Sequential filtration may also be combined with ultracentrifugation methods and allows for exosome isolation with high purity and high functional integrity due to low manipulation forces [22].

### 3.3. Precipitation Techniques

Polymeric Precipitation. Exosome isolation by precipitation occurs by introducing a polymer, typically poly-ethylene glycol (PEG), into the solution containing the exosomes [19]. When this occurs, the polymer binds to the water molecules and saturates the solution, forcing the other particles, including the exosomes, out of the solution [7]. Once precipitation occurs, the particles can be further isolated by either low speed centrifugation (1500× *g*) or by filtration [7]. Pretreatment of the sample by filtration and/or ultracentrifugation may be necessary to reduce contamination, as this technique precipitates not only exosomes but also other extracellular vesicles, proteins, and protein aggregates [20]. Isolation by precipitation is a fairly simple technique, which can be used with a variety of sample volumes, that provides a high exosome yield [23].

### 3.4. Immunoaffinity Capture-Based Techniques

Enzyme-Linked Immunosorbent Assay (ELISA). The proteins and receptors on the membranes of exosomes allow for antibodies to be used to develop immunoaffinity capture-based techniques [19]. In an enzyme-linked immunosorbent assay (ELISA), an antibody against a specific antigen is immobilized onto a surface, typically a well in a 96-well plate [19]. In the case of exosome isolation, the exosome expresses the target antigens and therefore becomes immobilized onto the well which contains the antibody, via antibody-antigen interactions [19,22]. The contents of the sample that are not exosomes are not immobilized onto the surface, as they do not contain the target antigen, and are then washed away [19]. The exosomes left behind may then be detected using a secondary antibody containing a detection substrate [19]. The surface markers used for immunoaffinity-based isolations such as ELISA include the exosome surface proteins CD9, CD41, CD63, and CD81, for which antibodies are developed [7]. This technique may be used to isolate and enrich exosomes present in serum, urine, and plasma that has an exosome yield comparable to isolation by ultracentrifugation, while using a smaller sample volume [7,24]. However, because of the small volume capacity of the 96-well plates often used in this technique, only a small amount sample can be used, resulting in a fairly low exosome yield [23].

Magneto-Immunocapture. The magneto-immunocapture technique developed by Zarovni et al. uses a biotinylated antibody attached to the surface of streptavidin-coated magnetic beads [19,26]. The antibody-coated beads are incubated in the sample containing the exosomes which carry the target antigen [19]. The exosomes then bind to the beads and can be separated from the solution using a magnet. In addition to using the antibodies against the exosome surface markers used in ELISA, some magneto-immunocapture techniques have immobilized the Tim4 protein onto the magnetic beads, as this protein is able to bind to phosphatidylserine located on the surface of exosomes [22]. This method is quick, easy to use, and has shown to have a comparable or better exosome yield compared to ultracentrifugation [22]. Additionally, this method can be used with any size sample volume and does not result in any damage to the exosomes’ structure or biological function [19].

### 3.5. Microfluidics-Based Techniques

Developments in microfluidics have provided new methods for exosome isolation, using platforms and channels that process small amounts of fluids, on the microliter to picoliter scale [7]. Many of the devices used in microfluidics are made using polydimethylsiloxane (PDMS), which is biocompatible and optically transparent and therefore very useful in device fabrication [7]. Microfluidics devices are able to isolate exosomes on the basis of size, density, and immunoaffinity, as in the methods already mentioned, and can also use newer sorting mechanisms via acoustic, electrophoretic, and electromagnetic manipulation [22]. The ability to use small sample volumes with this technique allows for the high purity of the isolated exosomes while also reducing the cost and time consumed in the process [7].

### 3.6. Commercial Kits Used in Cervical and Ovarian Cancer Research

As previously mentioned, commercial kits are available for exosome isolation that may employ at least one of the techniques already discussed. Two of these kits, Total Exosome Isolation by Invitrogen and ExoQuick-TC by Systems Biosciences, have been used successfully in research pertaining to cervical and ovarian cancer, in which the isolation of exosomes for further study was necessary. Invitrogen uses the isolation principle of immunoprecipitation, while ExoQuick-TC relies on density centrifugation. Both have been shown to be effective for exosome isolation in ovarian cancer research [23,25,27,28].

## 4. The Clinical Applications of Exosomes in Gynecologic Malignancies

The advancement of lab technologies and techniques has opened the door to incredible discoveries pertaining to the role of exosomes in clinical medicine. This includes novel approaches to therapeutics and diagnostics within the field of oncology. Although there is much left to discover, there has already been some headway in both diagnostic and therapeutic applications of exosomes in gynecologic malignancies. Figure 4 illustrates the clinical future of exosomes in cancers of gynecologic organs.

### 4.1. Diagnostics

In ovarian cancer, high mortality is largely due to diagnosis in the late stage due to the lack of screening or tools for early diagnosis. Therefore, there is a great interest in identifying early biomarkers in ovarian cancer. A strong correlation between the molecular profiles of tumor cells and their derived exosomes has been established [29], indicating that circulating tumor-derived exosomes have diagnostic promise. The number of tumor-derived exosomal markers for ovarian cancer continues to become more extensive. Studies over the last ten years have identified a number of proteins (CA-125, epithelial cell surface antigen (EpCAM), CD24) and micro RNAs (miR-141, miR-21, miR-200a, miR-200b, miR-200c, miR-214, miR-205 and miR-203) carried as exosomal cargo and used to differentiate patients with benign from malignant disease, and correlate the early vs. late stages [10]. A study in 2019 identified that an exosome biomarker panel of HER2, EGFR, FRα, CA-125, EpCAM, CD24, CD9 and CD63 from 10 microliters of serum was able to discriminate cancer patient groups from benign subjects, and also successfully distinguished early and late-stage ovarian cancer in a cohort of 20 patient samples [30]. Moreover, the protein CLDN4 has been identified in circulating exosomes derived from ovarian cancer cells, allowing the possibility of early ovarian cancer diagnosis [4]. The utilization of exosomes in such a way would be integral to improving patient outcomes in ovarian cancer. Exosome capturing devices isolate specific exosomes, allowing for analysis of their contents. To date, three such devices have undergone varying levels of in vivo and ex vivo studies in ovarian cancer: PDMS chips, nano-plasmonic exosome sensors, and hemopurifiers [16]. These technologies show promise for the clinical utilization of exosomes in both screening and diagnostic testing.

In the realm of cervical cancer, understanding the integral role HPV plays has led to impressive developments in cervical cancer detection and prevention. Cervical cytology to identify precursor cancer cells, combined with information regarding HPV status, is the mainstay of early cervical cancer risk stratification and intervention. However, these methods are invasive, and the results are subject to variations in interpretation. For this reason, attention has turned to the use of exosomes as biomarkers in cervical cancer diagnosis. From plasma samples, the exosomal microRNAs miR-30d-5p and let-7d-3p have been identified as potential biomarkers for early cervical cancer detection [31].

### 4.2. Therapeutics

With increased understanding of exosomes in the communication required for tumor growth and metastasis, they have become an area of interest in cancer therapeutics. Exploiting the affinity of some exosomes for cancer cells by usning them as drug delivery vehicles has had promising outcomes in lung, kidney, and breast cancer cells [9,32]. In one lung and kidney cancer study, exosome-delivered Paclitaxel, a cytotoxic chemotherapy, was found to remain localized in a lung carcinoma model and be fifty times more cytotoxic to multi-drug resistant canine renal carcinoma cells [20].

Peritoneal dissemination and chemoresistance are two of the most challenging properties of ovarian cancer, making it hard to treat. Exosomes from cisplatin-resistant ovarian cancer cells had higher levels of cisplatin export transporters [33], illustrating how exosomes have an important role in chemoresistance mechanisms in the primary tumor. Exosomes also have the potential to be utilized in detecting treatment response. The exosome cargo proteins TGF-B11 and MAGE3/61 have demonstrated potential roles as biomarkers in monitoring the response to ovarian cancer treatment [4]. Likewise, identifying and inhibiting chemoresistance through microRNA miR-21-3p [4] may allow for more effective treatment using already well-established modalities.

Exosomes derived from adipose mesenchymal stem cells have been utilized to inhibit the cell proliferation of ovarian cancer cells [9]. This suggests a potential therapeutic application in disseminated ovarian cancer cells. Another therapeutic approach is inhibiting exosomes at the level of biogenesis and secretion to inhibit the cellular communication required for metastasis. Five such inhibitors: tipifarnib, neticonazole, climbazole, ketoconazole, and triadimenol, have been identified in in vitro studies, but their in vivo efficacy have not been studied [34]. GW4869 has demonstrably inhibited exosome secretion from ovarian cancer cells, leading to decreased ovarian cancer cell invasion [19], a promising step in ovarian cancer therapeutics. There is promising evidence for a role of exosomes in immunotherapy in other cancers [32]. However, evidence in gynecologic malignancies to-date remains in the early stages. Table 2 below summarizes the identified markers and their clinical potential in gynecologic malignancies.

## 5. Conclusions and Future Directions

The developments in identifying and understanding exosomes and their cargo is a monumental step in cancer biology. Gynecologic malignancies as a group vary significantly in initial clinical presentation, degree and pattern of spread, and treatment modalities. Although advances in technology and the knowledge gained of these diseases in the past decades have greatly improved diagnosis, treatment, and ultimately survival for patients with gynecologic malignancies, there are still many unanswered questions, and room for improvements in diagnosis and treatment.

In the realm of ovarian cancer, lack of screening and early diagnosis options is a significant disadvantage for the majority of patients, in addition to being is a major contributing factor to the high mortality rate. Knowing that tumor-derived exosomes carry molecular cargo correlated with the primary tumor cell yields a promising potential for exosomes to serve as biomarkers that could aid in early detection of cancer.

Ovarian cancer-derived exosomes may also carry biomarkers such as TGF-B11 and MAGE3/61 to identify response to treatment in a timely manner. Clinically, this knowledge may permit timely changes in treatment before the development of diffuse disease progression and further medical complications. The rapid dissemination of epithelial ovarian cancer, the most common of ovarian cancers, creates many treatment challenges. Understanding the vital role that exosomes play in this metastatic pattern has opened the door to treatment strategies targeting inhibition of their actions. For example, it has been suggested that the exosomal microRNAs miR-200b and 200c may contribute to tumor progression, therefore their detection in serum has diagnostic potential in recognizing treatment failure [4]. Additional studies continue to contribute to the field, and show potential in exosome-based diagnostic and therapeutic strategies in ovarian cancer. The field, while encouraging, it is still nascent, and exosome research as it relates to ovarian cancer is worthwhile to continue exploring.

The promising evidence of exosomes in cervical cancer and endometrial cancer, the most common gynecologic malignancies in the world and in the US respectively [21], can also be appreciated. While breakthroughs in cervical cancer prevention and screening through the Papanicolaou test have led to a significant improvement in cervical cancer diagnosis and survival [31] in developed nations, worldwide screening is not as accessible. One reason is the invasive nature of cervical cancer screening, requiring a cervical swab for sampling in a physician’s office. The utility of serum exosomes in serving as biomarkers to screen for or detect early cervical cancer holds promise to bridge the gap between the developed countries and the remainder of the world, since blood sampling could be performed in any community health center with no need for a specialized physician. Specifically, the identification of microRNAs let-7d-3p and miR-30d-5p in circulating cervical tumor derived exosomes in serum holds promise for the early diagnosis of cervical cancer [31] that may be more accessible globally than methods currently in place. Additionally, as in the case of ovarian cancer, the tumor-derived exosomes themselves may serve as targets for cervical cancer therapy. Targeting exosomal miR-221-3p to inhibit tumor angiogenesis or lymphogenic spread has been identified as a viable therapeutic strategy [12,14] that could decrease the progression of cervical cancer, ultimately improving patient outcomes. This highlights the opportunity to identify additional exosomal microRNA targets that could be used both for diagnosis and as target for cervical cancer therapy.

Similar to cervical cancer, clinical prognosis is significantly worse in cases of advanced disease progression in uterine cancer. The understanding of exosomes in uterine cancer is only just beginning. For example, it has been identified that preventing the downregulation of the tumor suppressor miR-148b within exosomes derived from endometrial cancer-associated fibroblasts is a potential therapy that inhibits endometrial cancer progression [18]. As with other gynecologic malignancies, exosomes are likely to be integral in the disease process of uterine cancer, carrying a role in early diagnostics, treatment monitoring, and prevention of metastasis. More research remains to be performed regarding the diagnostic and therapeutic interventions that can be developed from understanding exosomes in uterine cancer.

Over the years, the diagnosis and treatment standards in all gynecologic malignancies have evolved greatly, but there still remain large gaps in basic knowledge of the disease processes and clinical management of these diseases. Exosomes may be a key component to solving some of the unanswered critical questions in the disease onset, progression, and treatment, and closing the gaps so as to improve the outcomes of patients with gynecologic cancers around the world.

## Figures and Tables

**Figure 1 cancers-14-00158-f001:**
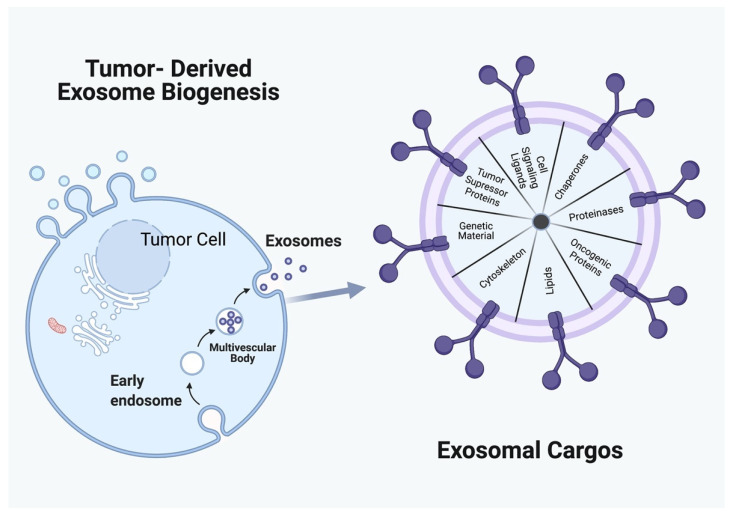
Tumor-derived exosome biogenesis and cargo. Created with BioRender.com (accessed on 17 December 2021).

**Figure 2 cancers-14-00158-f002:**
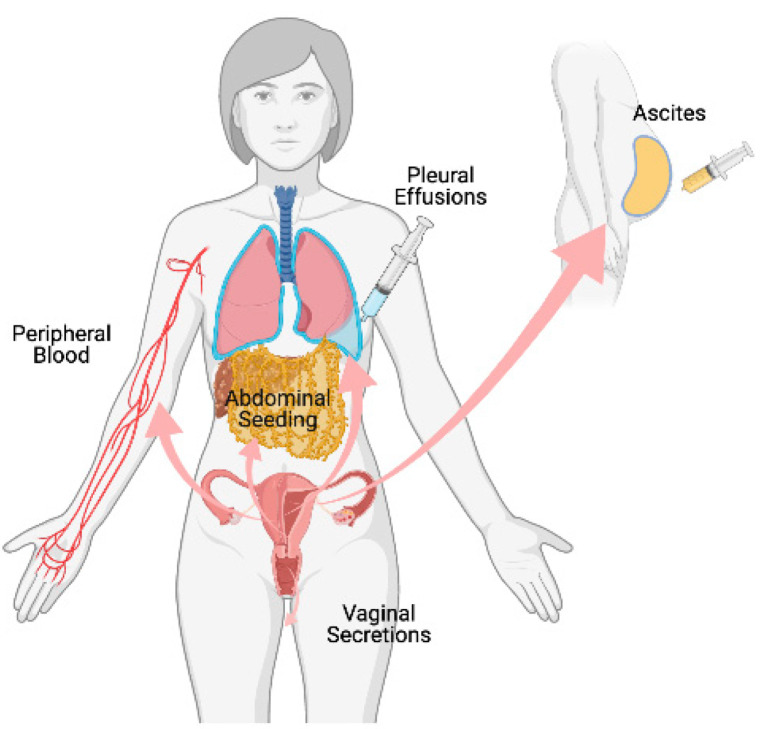
Gynecologic tumor-derived exosome spread in creating tumor microenvironments. Created with BioRender.com (accessed on 19 October 2021).

**Figure 3 cancers-14-00158-f003:**
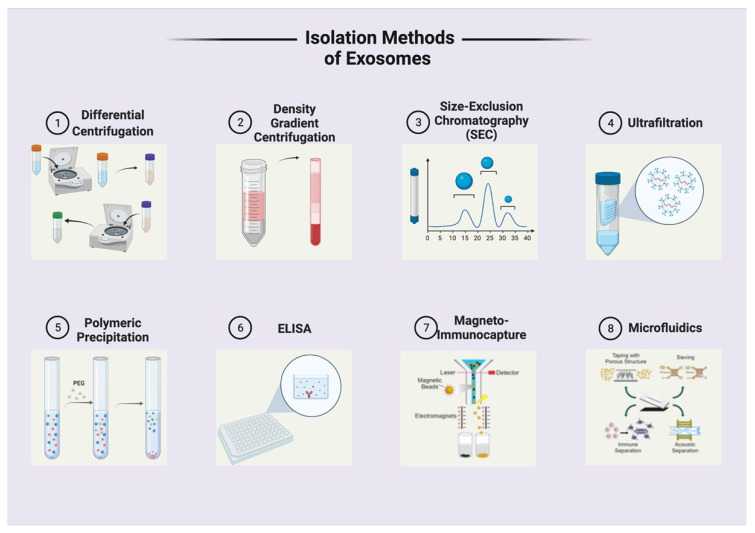
Exosome isolation techniques. Created with BioRender.com (accessed on 18 December 2021).

**Figure 4 cancers-14-00158-f004:**
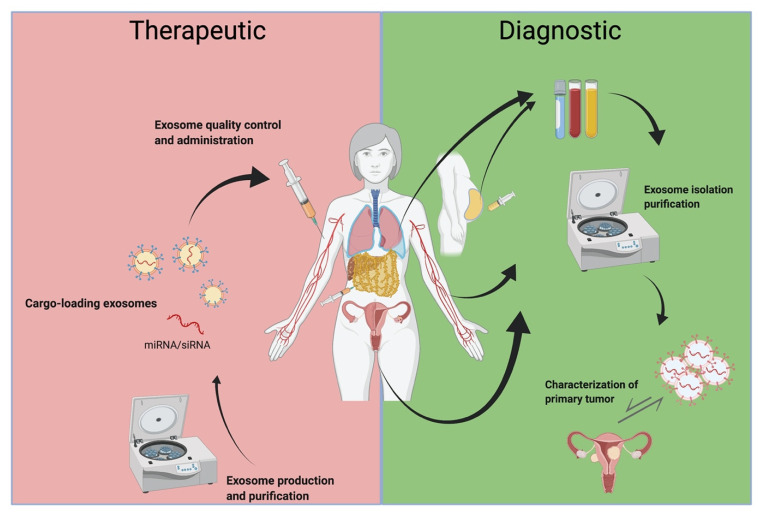
Clinical utility of exosomes in gynecologic malignancies. Created with BioRender.com (accessed on 20 October 2020).

**Table 1 cancers-14-00158-t001:** Summary of Exosome Isolation Techniques.

Isolation Technique	Isolation Principle	Protocol Overview	Advantages	Limitations
Differential Centrifugation	Density	Multiple centrifugation rounds at increasing speeds; each round eliminates a component, exosomes collected at the end	Simple protocol requiring little technical expertise. Can be used with a large sample volume	Low exosome yield and recovery, time consuming, may alter exosome structure and cause damage
Density Gradient Centrifugation	Density	Ultracentrifugation combined with the use of a density gradient medium	Higher purity exosomes compared to differential centrifugation	Time consuming, narrow loading area, may alter exosome structure and cause damage
Size-Exclusion Chromatography	Size	Use of a column containing porous beads which captures small particles and allow large particles to flow through. Exosomes eluted last using a buffer	Relatively low cost, high purity, no albumin contamination	Can only take small sample volume, low yield
Ultrafiltration	Size	Use of a nanoporous membrane to filter particles based on size. Exosomes are trapped through the pores	Higher exosome purity and less time consuming that ultracentrifugation	Exosome structure deformation from use of force, reduced recovery rate due to exosome binding onto the membrane
Sequential Filtration	Size	A series of filtration steps using different sized filters. Exosomes collected with the final filtration	High purity, high functional integrity due to low manipulation forces	Reduced recovery rate due to exosome binding onto the membrane
Polymeric Precipitation	Precipitation	Addition of a polymer, PEG, into the solution to precipitate the exosomes out of solution. Low speed centrifugation further isolates exosomes	Relatively simple technique, may be used with a variety of sample volumes, high exosome yield	Pretreatment required to avoid contamination
Enzyme-Linked Immunosorbent Assay (ELISA)	Immunoaffinity	Immobilization of an antibody onto a surface which binds to the antigen that exists on the exosome surface.	High exosome purity, uses a smaller sample volume, can isolate from serum, urine, and plasma	Low exosome yield
Magneto-Immunocapture	Immunoaffinity	Antibody added to streptavidin-coated magnetic beads. Exosome binds to the brands, which are isolated using a magnet	Quick and easy to use, high exosome yield, can be used with large or small sample volumes	
Microfluidics	Size, Density, Immunoaffinity, Electrophoresis, Magnetophoresis, Acoustohphoresis	Fabrication of a device that can isolate exosomes from a small volume of fluid. May use any of the established or new isolation principles.	High purity, reduced cost, little time consumed	

**Table 2 cancers-14-00158-t002:** Exosome markers identified in gynecologic malignancies.

Exosome Marker	Marker Type	Origin	Biologic Function	Potential Clinical Application
CLDN4^1^	Protein	Ovarian cancer	Exosome expression in circulation	Diagnostic: Early Detection
TGF-B1^1^	Protein	Ovarian cancer	Exosome expression in circulation	Monitoring therapeutic response
MAGE3/6^1^	Protein	Ovarian cancer	Exosome expression in circulation	Monitoring therapeutic response
miR-200b^1^	Micro RNA	Ovarian cancer	Tumor progression	Therapeutic: Target for inhibiting tumor progression
miR-200c^1^	Micro RNA	Ovarian cancer	Tumor progression	Therapeutic: Target for inhibiting tumor progression
miR-21-3p^1^	Mirco RNA	Ovarian cancer	Target: NAV3 gene → suppress apoptosis → Chemoresistance	Therapeutic: Target for inhibiting chemo resistance
let-7d-3p^16^	Mirco RNA	Cervical cancer	Exosome expression in circulation	Diagnostic: Early detection
miR-30d-5p^16^	Micro RNA	Cervical cancer	Exosome expression in circulation	Diagnostic: Early detection
miR-221-3p^14, 17^	Mirco RNA	Cervical cancer	Angiogenesis, Lymphogenesis	Therapeutic: Target for tumor growth and lymph node metastasis inhibition
miR- 148b^6^	Mircro RNA	Endometrial cancer associated fibroblasts (CAF)	Endometrial cancer growth inhibition	Therapeutic: Target for endometrial cancer growth inhibition

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
