# Peer review of "Understanding the Role and Clinical Applications of Exosomes in Gynecologic Malignancies: A Review of the Current Literature"

_cancers, 2021, doi:10.3390/cancers14010158_

Round 1

Reviewer 1 Report

1.This is a pertinent research work conducted by the authors and the review would benefit the research community in the field of gynecologic malignancies.

2.The role of exosomes in gynecologic cancers is very well described and all the major cancer types have been covered.

3. An important aspect of the review is the methodology of the isolation of the exosomes which are covered in detail would be useful for researchers working in the field.

4.Summarization of the table for exosome markers for the identification in gynecologic malignancies is very well done and provides relevant information.

#5 The english language needs to be checked in certain places.

Reviewer 2 Report

The authors have written a timely and useful review of the biology and potential diagnostic relevance of exosomes in ovarian, cervical, and uterine cancer. The review is well-organized, well-written, and informative. The paper would be improved with the following:

The introduction (Section 1) contains several statements that should be supported with references, particularly the first paragraph. Additionally, the references in the introduction may be misnumbered.

In Figure 1, "cell signaling ligands" is used twice. Is that intended?

Please capitalize "United States" in the first sentence of Section 2C.

Please correct the grammar in the first sentence of Section 3.

Figure 3 should be improved in several ways.

  • There are two separate boxes for 'immunoaffinity based techniques'. These should be combined or otherwise differentiated with different titles.
  • There is only one size-based method summarized. The text describes several filtration methods and these should be summarized in Figure 3 alongside chromatography.
  • Precipitation is not a necessary step in size-based isolation methods.
  • Ultracentrifugation (100,000 xg) is not a necessary step in either size-based isolation or density gradient centrifugation.

In Section 3F, please clarify that ExoQuick is not an immunoisolation technique. ExoQuick relies on density centrifugation.

In Section 4B, it is unclear what the devices mentioned have to do with a therapeutic approach to cancer. “Another technology utilizing exosomes has been exosome capturing devices, which bind specific exosomes containing antigens, and remove them. Three such devices have been studied in ovarian cancer, PDMS chip, Nano-plasmonic exosome sensor, and Hemopurifier with varying degrees of in vivo and ex vivo studies to  date [12].” Are these devices 'therapeutic', or are they better described as diagnostic or prognostic technologies?

The sentence “There is promising evidence for exosome-based immunotherapy in other cancers, however evidence in gynecologic malignancies to-date remains in the early stages.” requires a reference.

The second paragraph of Section 5 appears to summarize the conclusions of another review on exosomes in ovarian cancer, rather than providing the authors' conclusions or future perspectives. The proteins TGF-B11 and MAGE3/6 only appear here and in the the summary table. If they are important for ovarian cancer or exosomes in general, they should be more thoroughly described in the main body of the text.

Author Response

Thank you for your feedback. Revisions have been made as requested. Please see the attachment. 
